# Probabilistic and Geometric Depth: Detecting Objects in Perspective

**Tai Wang**[1,2]    **Xinge Zhu**[1]    **Jiangmiao Pang**[1,2*]    **Dahua Lin**[1,2,3]

[1]CUHK-SenseTime Joint Lab, The Chinese University of Hong Kong

[2]Shanghai AI Laboratory    [3]Centre of Perceptual and Interactive Intelligence

{wt019, zx018, dhlin}@ie.cuhk.edu.hk, pangjiangmiao@gmail.com

**Abstract:** 3D object detection is an important capability needed in various practical applications such as driver assistance systems. Monocular 3D detection, as a representative general setting among image-based approaches, provides a more economical solution than conventional settings relying on LiDARs but still yields unsatisfactory results. This paper first presents a systematic study on this problem. We observe that the current monocular 3D detection can be simplified as an instance depth estimation problem: The inaccurate instance depth *blocks* all the other 3D attribute predictions from improving the overall detection performance. Moreover, recent methods directly estimate the depth based on isolated instances or pixels while ignoring the geometric relations across different objects. To this end, we construct *geometric* relation graphs across predicted objects and use the graph to facilitate depth estimation. As the preliminary depth estimation of each instance is usually inaccurate in this ill-posed setting, we incorporate a *probabilistic* representation to capture the uncertainty. It provides an important indicator to identify confident predictions and further guide the depth propagation. Despite the simplicity of the basic idea, our method, PGD, obtains significant improvements on KITTI and nuScenes benchmarks, achieving 1st place out of all monocular vision-only methods while still maintaining real-time efficiency. Code and models will be released at https://github.com/open-mmlab/mmdetection3d.

**Keywords:** Probabilistic and Geometric Depth, Monocular 3D Detection

## 1 Introduction

3D object detection is an essential task for many robotic systems such as autonomous vehicles. Recent advanced methods in this field typically resort to various sensors, such as LiDAR [1, 2, 3, 4, 5, 6], Radar [7], binocular vision [8, 9], or their combinations for accurate depth information. Nevertheless, these perceptual systems are complicated, expensive, and difficult to maintain in complex environments. In contrast, monocular 3D detection, a setting that aims at perceiveing 3D objects from 2D monocular images, has drawn increasing attention due to its low costs. However, as the depth information is not directly manifest in the input, this task is inherently ill-posed, making the problem particularly challenging.

This paper starts from a systematic study about this problem on two authoritative benchmarks in a quantitative way. Although we already knew the depth information is critical to this task, the study surprisingly shows that inaccurate depth estimation blocks all the other localization predictions from improving the final results. As instance depth has shown to be the bottleneck, we can simplify monocular 3D detection as an instance depth estimation problem to tackle it essentially.

Previous methods [10, 11] first use an extra cumbersome depth estimation model to complement 2D detectors on depth information. The following methods [12, 13, 14] simplify the frameworks by directly regarding depth as one dimension of the 3D localization task. However, they still use simple methods that estimate depth from isolated instances or pixels in a regression manner. We observe that aside from each object itself, other objects are co-existing in an image and the geometric relations across them can be valuable constraints to guarantee accurate estimation.

Motivated by these observations, we propose Probabilistic and Geometric Depth (PGD) that jointly leverages probabilistic depth uncertainty and geometric relationships across co-existed objects for accurate depth estimation. Specifically, as the preliminary depth estimation of each instance is usually inaccurate in this ill-posed setting, we incorporate a probabilistic representation to capture the uncertainty of the estimated depth. We first bucket the depth values into a set of intervals and calculate the depth by the expectation of the distribution (Fig. 1(a)). The average of top-k confidence scores from the distribution is taken as the uncertainty of the depth. To model the geometric relations,

---

*Corresponding author

5th Conference on Robot Learning (CoRL 2021), London, UK.

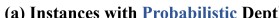
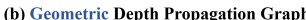
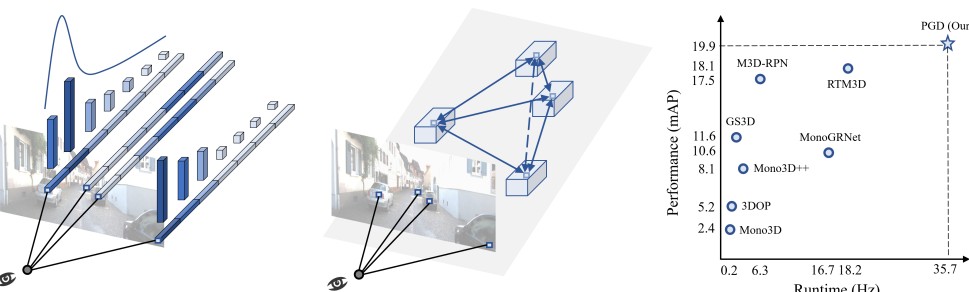

Figure 1: In this paper, to tackle the dominating depth estimation problem in the monocular 3D detection, we first (a) predict the depth of each instance with a probabilistic representation to capture the uncertainty, and (b) further construct a geometric relation graph to enhance the estimation from contextual connections. (c) The proposed method outperforms the other work in terms of both performance and speed significantly on the KITTI 3D car detection benchmark.

we further construct a depth propagation graph to enhance the estimations with their contextual relationship. The uncertainty of each instance depth provides useful guidance for the propagation therein. Benefiting from this overall scheme, we can easily identify the predictions with higher confidence, and more importantly, estimate their depths more accurately with the graph-based synergistic mechanism.

We implement the methods on a simple monocular 3D object detector FCOS3D [15]. Despite the simplicity of the basic idea, our PGD results in significant improvements on KITTI [16] and nuScenes [17] with different benchmark settings and evaluation metrics. It achieves 1st place out of all monocular vision-only methods while still maintaining real-time efficiency. The simple yet effective method proves that with only designs tailored to depth, a 2D detector can be capable of detecting objects in perspective.

## 2 Related Work

**2D Object Detection** According to the base of initial guesses, modern 2D detection methods can be divided into two branches, anchor-based and anchor-free. Anchor-based methods [18, 19, 20, 21] benefit from the predefined anchors in terms of much easier regression, while anchor-free methods [22, 23, 24, 25, 26] do not need complicated prior settings and thus have better universality. For simplicity, we take FCOS3D [15], the 3D adapted version of FCOS [24], as the baseline considering its capability of handling overlapped ground truths and scale variance problem.

**Monocular Depth Estimation** Monocular depth estimation is also a challenging ill-posed problem like monocular 3D detection. It aims at predicting dense and global depth field at pixel level given an RGB image. Early works [27, 28] predict depth from hand-crafted features with non-parametric optimization methods. With the rapid progress of CNNs, fully supervised methods [29, 30, 31], self-supervised methods based on stereo pairs [32, 33] and monocular videos [34, 35] gradually emerged. Although this problem has been explored for a long time, there are very few works [36] studying it in a specific task, like detection, where the dense depth supervision is always not guaranteed and we only care about the accuracy of instance depth instead of the global depth field.

As for the reformulation of depth learning problems, there are a few attempts in this field. For example, DORN [37] recasts the depth learning problem as ordinal regression and proposes a spacing-increasing discretization (SID) strategy to improve network training and reduce computations. It is similar to the underlying idea of our probabilistic representation for uncertainty modeling while different in terms of motivation and design details.

**Monocular 3D Object Detection** Monocular 3D detection is more complicated than the 2D case. The underlying problem is the inconsistency of input 2D data modal and the output 3D predictions.

*Methods involving sub-networks* Earlier work uses sub-networks to assist 3D detection. 3DOP [38] and MLFusion [10] use a depth estimation network while Deep3DBox [39] uses a 2D object detector. They rely on the design and performance of these sub-networks, even external data and pre-trained models, which makes the training inconvenient and introduces additional system complexity.

*Transform to 3D representation* Another category is to convert the RGB input to 3D representations like OFTNet [40] and Pseudo-Lidar [11]. Although these methods have shown promising

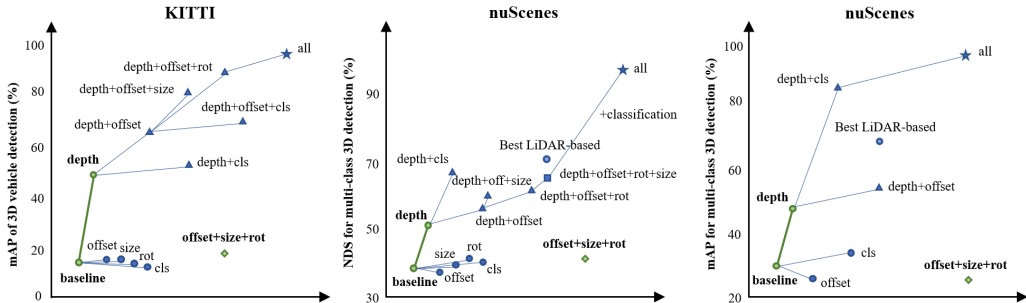

Figure 2: Oracle analyses with different datasets and metrics. From left to right: 3D IoU based mAP on KITTI, NuScenes Detection Score (NDS) and distance-based mAP on nuScenes. We replace our predictions with ground truth values step by step and observe the performance improvements. It can be seen that an accurate depth can bring significant performance improvement (green lines), and only with accurate depth can the improvements brought by other oracles be realized.

performance, they actually rely on dense depth labels and hence are not regarded as pure monocular approaches. There are also domain gaps between different depth sensors, making them hard to generalize smoothly to a new practical setting. Furthermore, the efficiency of processing a large number of point clouds is also a significant issue to deal with in practical applications.

*End-to-end designs like 2D detection*    Recent work notices these drawbacks, and end-to-end frameworks are thus proposed. M3D-RPN [13] implements a single-stage multi-class detector with an end-to-end region proposal network and depth-aware convolution. SS3D [41] proposes to detect 2D key points and further predicts object characteristics with uncertainties. MonoDIS [12] introduces a disentangling loss to reduce the instability of the training procedure. Some of them still have multiple training stages or post-optimization phases. In addition, they all follow anchor-based manners, and thus the consistency of 2D and 3D anchors is needed to be determined. In contrast, anchor-free methods [26, 14, 42, 15] do not need to make statistics on the given data and have better generalized ability to more various classes or different intrinsic settings, so we choose to follow this paradigm.

Nevertheless, all of these works rarely have customized designs for instance depth estimation in particular, and only take it as one common regression target for isolated points or instances. It actually hinders the breakthrough of this problem, which will be discussed in our quantitative study and specifically addressed in our approach.

## 3    Preliminary and Motivating Study

In this section, we aim at making an in-depth quantitative error analysis on top of a basic adapted monocular 3D detector to investigate the key challenge in the specific 3D detection setting.

Typically, conventional 2D detection expects the model to predict 2D bounding boxes and category labels for each object of interest, while a monocular 3D detector needs to predict 7-DoF 3D boxes given the same input. From this perspective of problem formulation, the main difference lies on the regression targets. An intuitive reason for the much worse performance of monocular 3D detection compared to 2D is that there exist much more difficult targets to regress in the *localization*. Hence, we choose a simple detector FCOS3D [15] to study the specific problem, which keeps the well-developed designs for 2D feature extraction and is adapted for this 3D task with only basic designs for specific 3D detection targets. As shown in the left part of Fig. 3, there are overall two branches for classification and localization respectively. Formally, for the regression branch, the detector predicts 3D attributes, including offsets $\Delta$x, $\Delta$y to the projected 3D center, depths $d$, 3D size $w^{3D}$, $l^{3D}$, $h^{3D}$, sin value of rotation $\theta$, direction class $C_\theta$, center-ness $c$, and distances to four sides of 2D boxes $l$, $r$, $t$, $b$, for each location on the output dense map. We further equip it with a basic consistency loss between 3D and 2D localization, which will be detailed in supplemental materials.

On this basis, we apply this baseline on two representative benchmarks, KITTI and nuScenes, and replace the predictions with ground truths step by step to identify the performance bottleneck (Fig. 2). Unexpectedly, the *inaccurate depth* blocks *all* the other sub-task predictions from improving the overall detection performance, on both datasets under different metrics. Hence, current monocular 3D detection, especially 3D localization, can be reduced to the dominating instance depth estimation problem to a great extent, which will be the focus of our method to be presented next. See more details about the explanation of oracle analyses in supplemental materials.

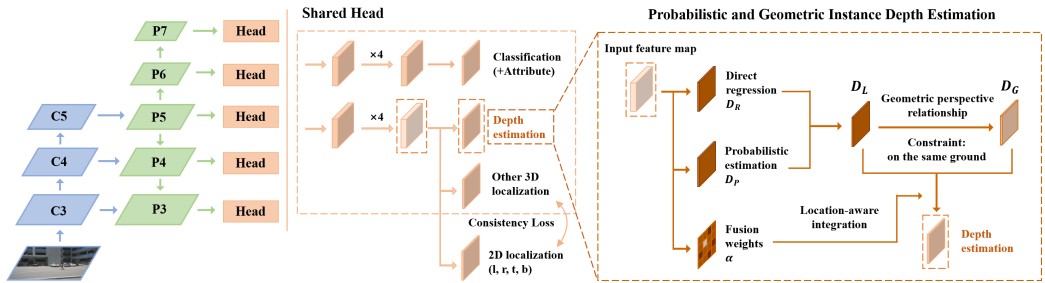

Figure 3: An overview of our framework. We start from a basic monocular 3D detector, FCOS3D, while focus on tackling the difficulty of instance depth estimation with our proposed customized module in the head. With the feature map from the regression branch as the input, we first introduce a branch for probabilistic depth estimation to model the uncertainty, then derive the geometric depth with the depth propagation graph and finally integrate them to get the final depth prediction.

# 4 Our Approach

Given images collected from similar cameras, previous work typically resorts to direct regression for instance depth estimation and expects the model to directly learn that objects with certain appearances and sizes always exist at locations with certain depths. Our baseline also follows this way. However, it is hard to learn due to the large variance and also obviously not enough for the accuracy needed in 3D detection. Given the inherent downside of hard regression for isolated points, in our approach, we aim at constructing an uncertainty-aware depth propagation graph to enhance the estimation from contextual connections among instances. Next, we will first elaborate on the adopted probabilistic representation and technical details of the constructed geometric graph, and finally present how we integrate these obtained depth estimations.

## 4.1 Uncertainty Modeling with Probabilistic Representation

For a one-stage detector, a general design for direct depth estimation is a small head along the regression branch expected to output a dense depth map. Formally, as shown in Fig. 3, suppose the input feature map has shape $H \times W$, then the direct depth regression output [2] can be denoted as $D_R \in \mathbb{R}^{H \times W}$. On this basis, to establish an effective depth propagation mechanism, modeling the uncertainty of depth estimation for each instance is an important preliminary, which can provide useful guidance for weighing the propagation among instances. We adopt a simple yet effective probabilistic representation to achieve this: Considering the depth value is continuous in a certain range, we uniformly quantize the depth interval into a set of discrete values and represent the prediction with the expectation of the distribution. Suppose the detection range is $0 \sim D_{max}$, the discretized unit is $U$, then we have $C = \lfloor D_{max}/U \rfloor + 1$ split points. Denote the set of points as a weight vector $\omega \in \mathbb{R}^C$, and then we introduce a new head parallel with direct regression to produce a probabilistic output map $D_{PM}$, which will be decoded with:

$$D_P = \omega^T softmax(D_{PM}) \tag{1}$$

where $D_P$ is the so-called probabilistic depth. It is equivalent to compute the expectation of the probabilistic distribution formed by $softmax(D_{PM})$. Apart from the $D_P$, we can further obtain the depth confidence score, denoted as $s^d \in S_D$, from the depth distribution of each instance. In practice, we take the average of top-2 confidence as the depth score for $U = 10m$. It will be multiplied by the center-ness and classification score as the final ranking criterion for predictions during inference.

Subsequently, we fuse $D_R$ and $D_P$ with the sigmoid response of a data-agnostic single parameter $\lambda$:

$$D_L = \sigma(\lambda)D_R + (1 - \sigma(\lambda))D_P \tag{2}$$

Here $D_L$ is regarded as a *local* depth estimation for each isolated instance, which together with the depth score derived from $D_{PM}$ serve as the foundation of constructing the depth propagation graph.

It is worth noting that our implementation is different from the typical way used in monocular depth estimation [37], which usually adopts a fine-grained quantization for the depth interval and further estimates the value with classification and residual regression. In comparison, our method is more memory-efficient, more straightforward for regressing continuous value, and provides a natural indicator for uncertainty estimation. Please refer to the supplemental for empirical results about comparison with other depth interval division methods.

---

[2]To make learning easier, the output of direct regression branch is applied an exponential transformation.

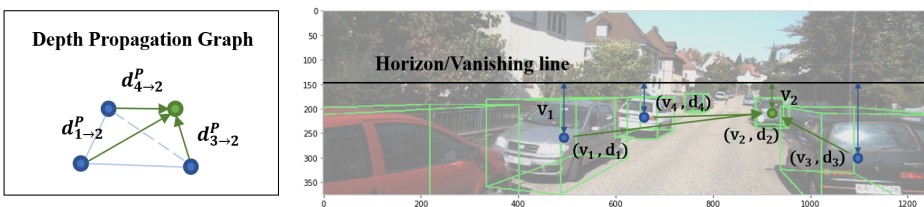

Figure 4: For objects hard to approximate depth accurately (like car 2), propagating (green arrows) reliable depth predictions from other objects (car 1, 3, 4) with perspective geometry can enhance the reasoning from the global context.

## 4.2 Depth Propagation from Perspective Geometry

With the depth prediction $D_L$ for isolated instances and their depth confidence scores $S_D$ for uncertainty estimation, we can further construct the propagation graph based on the contextual geometric relationship. Consider the typical driving scenarios: a general constraint can be leveraged, *i.e.*, almost all the objects are on the ground. Early in [43], Hoiem et al. utilized the scene projection with this constraint to put objects in the context of the overall 3D scene by modeling the relationship of different elements. Here, targeting the depth estimation problem, we instead propose a geometric depth propagation mechanism with consideration of interdependence between instances. Next, we will first derive the perspective relationship between two instances, and then present the details of the graph-based depth propagation scheme with edge pruning and gating.

**Perspective Relationship**  Consider in the general perspective projection, suppose the camera projection matrix $P$ is:

$$P = \begin{pmatrix} f & 0 & c_u & -fb_x \\ 0 & f & c_v & -fb_y \\ 0 & 0 & 1 & -fb_z \end{pmatrix} \tag{3}$$

where $f$ is the focal length, $c_u$ and $c_v$ are the vertical and horizon position of camera in the image, $b_x$, $b_y$ and $b_z$ denote the baseline with respect to the reference camera (non-zero in KITTI while zero in nuScenes). Note that we represent the focal length with a single $f$ considering most cameras share the same one for the $u$ and $v$ axis. Then a 3D point $\mathbf{x^{3D}} = (x, y, z, 1)^T$ in the camera coordinates can be projected to a point $\mathbf{x^{2D}} = (u', v', 1)^T$ in the image with:

$$d\mathbf{x_{2D}} = P\mathbf{x_{3D}} \tag{4}$$

To simplify the result, we replace $v'$ with $v + c_v$, then $v$ represents the distance to the horizon line (Down is the positive direction in the Fig. 4). Then we get:

$$vd = f(y - b_y + c_v b_z) \tag{5}$$

The relation for $u$ is similar. Considering the constraint that all the objects are on the ground, the bottom centers of objects always share the same $y$ (height in the camera coordinates), so we mainly consider this relation for $v$ next. Given two objects 1 and 2, the relationship between their center depths can be derived from Eqn. 5:

$$d_2 = \frac{v_1}{v_2}d_1 + \frac{f}{v_2}(y_2 - y_1) \approx \frac{v_1}{v_2}d_1 + \frac{f}{2v_2}(h_1^{3D} - h_2^{3D}) \triangleq d_{1\to 2}^P \tag{6}$$

with which we can predict $d_2$ given $d_1$ precisely with the height difference between 3D centers. Besides, we can also leverage an approximation of this relationship given the assumption that objects share the same bottom height, then $y_2 - y_1$ can be substituted by the difference of half heights of 3D boxes $\frac{1}{2}(h_1^{3D} - h_2^{3D})$, defined as $d_{1\to 2}^P$.

In this relation, when $h_1^{3D} = h_2^{3D}$, $v_1 d_1 = v_2 d_2$, which is easy to understand, *i.e.*, an object closer to vanishing line is farther away. It is a clear relationship connecting different instances but also yields errors. Suppose $|(y_2 - y_1) - \frac{1}{2}(h_1 - h_2)| = \delta$, the error of depth will be $\Delta d = \frac{f}{v_2}\delta$. When $\delta = 0.1m$, $v_2 = 50$ (pixels), $\Delta d$ can be about $1.5m$. Although it is acceptable for objects $30m$ away (corresponding with $v_2 = 50$), we also need a mechanism to avoid possible large errors. It consists of the edge pruning and gating scheme to be described next and the location-aware weight map to be mentioned in the Sec. 4.3.

**Graph-Based Depth Propagation**  With the pairwise perspective relationship, we can estimate the depth of any object from the cues of other objects. Then we can construct a dense directed graph with two bidirectional edges between any two objects representing the depth propagation

(Fig. 4). Formally, suppose we have $N$ predicted objects with indices from $\mathcal{P} = \{1, 2, ..., n\}$, we can estimate the depth of object $i$ given $d^P_{j \to i}$ for all the $j \in \mathcal{P}$, defined as the geometric depth $d^G_i \in D_G$. Considering the computational efficiency and possible large errors mentioned previously, we propose an edge pruning and gating scheme to improve the propagation graph. From our observation, the same category of nearby objects can well satisfy the "same ground" condition, so we select the following 3 most important factors to decide which edges are influential and reliable, including the depth confidence $s^d_j$, 2D distance score $s^{2D}_{ij}$, and classification similarity $s^{cls}_{ij}$. The latter two and the overall edge score $s^e_{j \to i}$ are computed as follows:

$$s^{2D}_{ij} = 1 - \frac{t^{2D}_{ij}}{t^{2D}_{max}}, \quad s^{cls}_{ij} = \frac{\boldsymbol{f_i} \cdot \boldsymbol{f_j}}{||\boldsymbol{f_i}||_2 ||\boldsymbol{f_j}||_2}, \quad s^e_{j \to i} = \frac{s^d_j \cdot s^{2D}_{ij} \cdot s^{cls}_{ij}}{\sum^k_{j=1} s^d_j \cdot s^{2D}_{ij} \cdot s^{cls}_{ij}} \tag{7}$$

where $t^{2D}_{ij}$ is the 2D distance between projected centers of object $i$ and $j$, $t^{2D}_{max}$ is set to the length of image diagonal, $\boldsymbol{f_i}$ and $\boldsymbol{f_j}$ are the output confidence vectors of two objects from classification branch and $k$ is the maximum number of edges to be kept after pruning (edges with top-$k$ scores are kept). The edge score is then used for gating so that each node attends its edges with their importance:

$$d^G_i = \sum^k_{j=1} s^e_{j \to i} d^P_{j \to i} \tag{8}$$

Note that obtaining the geometric depth map $D_G$ from this graph is free of learnable parameters. To avoid influencing the learning of other components, we cut off the gradients backpropagated from this computation and only focus on how to integrate $D_L$ and $D_G$, which will be discussed next.

### 4.3 Probabilistic and Geometric Depth Estimation

So far, we have obtained two depth predictions $D_L$ and $D_G$ from isolated and graph-based contextual estimations, respectively. Then we integrate these two complementary components in a learning manner. Unlike the data-agnostic single parameter used in the local estimation, integrating these two results should be more complex considering their flexible roles in various complicated cases. So we further introduce a branch to produce a location-aware weight map $\alpha \in \mathbb{R}^{H \times W}$ to fuse them (Fig. 3):

$$D = \sigma(\alpha) \circ D_L + (1 - \sigma(\alpha)) \circ D_G \tag{9}$$

The fused depth $D$ will replace the direct regressed $D_R$ in the baseline and trained with the common smooth L1 loss in the same end-to-end way. Note that adding intermediate supervisions empirically makes the training more stable but does not bring any performance gains.

## 5 Experiments

In this section, we present our experimental setup and implementation details, and then make the quantitative analysis on the KITTI and nuScenes dataset with details of both performance and efficiency. Finally, detailed ablation studies are conducted to show the efficacy of each component in our method. Refer to supplemental materials for more qualitative analysis.

### 5.1 Datasets & Evaluation Metrics

We evaluate our method on two datasets, KITTI [16] and nuScenes [17]. There are 7481/7518 samples for training/testing respectively on KITTI, and the training samples are generally divided into 3712/3769 samples as training/validation splits. We first validate our method on this popular benchmark. Nevertheless, the variety of scenes and categories is limited on KITTI, so we further test our approach on the large-scale nuScenes dataset. NuScenes consists of multi-modal data collected from 1000 scenes, including RGB images from 6 cameras, points from 5 Radars, and 1 LiDAR. It is split into 700/150/150 scenes for training/validation/testing. There are overall 1.4M annotated 3D bounding boxes from 10 categories. In addition, nuScenes uses different metrics, distance-based mAP and NDS, which can help evaluate our method from another perspective. See more explanations about metrics in supplemental materials.

### 5.2 Implementation Details

**Network Architectures**   As shown in the Fig. 3, our baseline framework basically follows the design of FCOS3D [15]. Given the input image, we utilize ResNet101 [44] as the feature extraction backbone followed by FPN [45] for generating multi-level predictions. Detection heads are shared among multi-level feature maps except that three scale factors are used to differentiate some of their

Table 1: Results on the KITTI validation dataset

| Methods | Venue | Extra Labels | Time | $AP_{BEV}$ IOU$\geq$ 0.7 | | | $AP_{3D}$ IOU$\geq$ 0.7 | | |
|---|---|---|---|---|---|---|---|---|---|
| | | | | Easy | Mod. | Hard | Easy | Mod. | Hard |
| Mono3D [47] | CVPR 2016 | Mask | 4.2s | 5.22 | 5.19 | 4.13 | 2.53 | 2.31 | 2.31 |
| 3DOP [38] | TPAMI 2017 | Stereo | 3s | 12.63 | 9.49 | 7.59 | 6.55 | 5.07 | 4.10 |
| MF3D [10] | CVPR 2018 | Dense Depth | - | 22.03 | 13.63 | 11.60 | 10.53 | 5.69 | 5.39 |
| Mono3D++ [48] | AAAI 2018 | Dense Depth+Shape | >0.6s | 16.70 | 11.50 | 10.10 | 10.60 | 7.90 | 5.70 |
| PL [11, 36] (AVOD) | CVPR 2019 | Dense Depth | - | 19.0 | 15.3 | 13.0 | 7.5 | 6.1 | 5.4 |
| ForeSeE [36] (AVOD) | AAAI 2020 | Dense Depth | - | 23.4 | 17.4 | 15.9 | 15.0 | 12.5 | 12.0 |
| Deep3DBox [39] | CVPR 2018 | None | - | 9.99 | 7.71 | 5.30 | 5.85 | 4.10 | 3.84 |
| MonoGRNet [49] | AAAI 2019 | None | 0.06s | - | - | - | 13.88 | 10.19 | 7.62 |
| FQNet [50] | CVPR 2019 | None | 3.33s | 9.50 | 8.02 | 7.71 | 5.98 | 5.50 | 4.75 |
| GS3D [51] | CVPR 2019 | None | 2.3s | - | - | - | 13.46 | 10.97 | 10.38 |
| M3D-RPN [13] | ICCV 2019 | None | 0.16s | 25.94 | 21.18 | 17.90 | 20.27 | 17.06 | 15.21 |
| MonoDIS [12] | ICCV 2019 | None | - | 24.26 | 18.43 | 16.95 | 18.05 | 14.98 | 13.42 |
| RTM3D [14] | ECCV 2020 | None | 0.055s | 25.56 | 22.12 | **20.91** | 20.77 | 16.86 | 16.63 |
| FCOS3D [15] | ICCVW 2021 | None | - | 18.16 | 14.02 | 13.85 | 13.90 | 11.61 | 10.98 |
| PGD (Ours) | - | None | **0.028s** | **30.56** | **23.67** | 20.84 | **24.35** | **18.34** | **16.90** |

Table 2: Results on the nuScenes dataset.

| Methods | Split | Modality | mAP | mATE | mASE | mAOE | mAVE | mAAE | NDS |
|---|---|---|---|---|---|---|---|---|---|
| PointPillars (Light) [2] | test | LiDAR | 0.305 | 0.517 | 0.290 | 0.500 | 0.316 | 0.368 | 0.453 |
| CenterFusion [7] | test | Cam. & Radar | 0.326 | 0.631 | 0.261 | 0.516 | 0.614 | 0.115 | 0.449 |
| CenterPoint v2 [52] | test | Cam. & LiDAR & Radar | **0.671** | 0.249 | 0.236 | 0.350 | 0.250 | 0.136 | **0.714** |
| LRM0 | test | Camera | 0.294 | 0.752 | 0.265 | 0.603 | 1.582 | 0.14 | 0.371 |
| MonoDIS [12] | test | Camera | 0.304 | 0.738 | 0.263 | 0.546 | 1.553 | 0.134 | 0.384 |
| CenterNet [26] | test | Camera | 0.338 | 0.658 | 0.255 | 0.629 | 1.629 | 0.142 | 0.4 |
| Noah CV Lab | test | Camera | 0.331 | 0.660 | 0.262 | 0.354 | 1.663 | 0.198 | 0.418 |
| FCOS3D [15] | test | Camera | 0.358 | 0.690 | 0.249 | 0.452 | 1.434 | 0.124 | 0.428 |
| PGD (Ours) | test | Camera | **0.386** | 0.626 | 0.245 | 0.451 | 1.509 | 0.127 | **0.448** |
| CenterNet [26] | val | Camera | 0.306 | 0.716 | 0.264 | 0.609 | 1.426 | 0.658 | 0.328 |
| FCOS3D [15] | val | Camera | 0.343 | 0.725 | 0.263 | 0.422 | 1.292 | 0.153 | 0.415 |
| PGD (Ours) | val | Camera | **0.369** | 0.683 | 0.260 | 0.439 | 1.268 | 0.185 | **0.428** |

final regressed results, including offsets, depths, and sizes, respectively. For the hyperparameters in the depth estimation module, $U$ is set to $10m$ and $k$ is set to 5. The overall framework is built on top of MMDetection3D [46]. Please refer to FCOS3D [15] and supplemental materials for the design of loss and other implementation details.

**Training Parameters** For all the experiments, we trained randomly initialized networks from scratch following end-to-end manners. Models are trained with SGD optimizer, in which gradient clip and warm-up policy are exploited with learning rate 0.001, number of warm-up iterations 500, warm-up ratio 0.33 and batch size 32/12 on 16/4 GTX 1080Ti GPUs for nuScenes/KITTI.

**Data Augmentation** We only implement image flip for augmentation, where offset and 2D targets are flipped for the 2D image while 3D boxes are transformed correspondingly in 3D space. No other augmentation (right image augmentation, cropping, resizing, *etc.*) methods are utilized.

### 5.3 Quantitative Analysis

We make quantitative analyses both on KITTI (Tab. 1 and 4, Fig. 1(c)) and much harder, less commonly validated nuScenes dataset (Tab. 2). It can be seen that our method achieves the state-of-the-art on both benchmarks with different settings and metrics while maintains outstanding speed.

We list part of early monocular methods with extra data or pre-trained models and recent image-only methods that have related results for comparison on the KITTI dataset. Only the results for car detection are compared here because the performance of small objects is always unstable due to their limited samples. Our framework based on the simple adapted FCOS3D achieves much better performance than others, especially considering M3D-RPN [13] and RTM3D [14] adopt stronger backbone and data augmentation. Furthermore, our method can run at the speed of 36Hz to achieve this, thanks to most of our modules not introducing extra computational costs to inference. It is an excellent trade-off between performance and efficiency.

Then for the nuScenes dataset, we also compare the results on the test set and validation set, respectively. On the test set, we first compared all the methods using RGB images as the input data. Our single model achieved the best performance among them with mAP 37.0% and NDS 43.2%, in which we particularly exceeded the previous best method more than 3% in terms of mAP. We also list benchmarks based on other data modality, including lightweight, real-time PointPillars [2] with LiDAR, CenterFusion [7] with RGB image and Radar, and CenterPoint [52] ensemble results with all the sensors. It can be seen that although our method has a certain gap with the high-performance CenterPoint, it even surpasses PointPillars and CenterFusion on mAP, which shows that this ill-posed

Table 3: Ablation study on KITTI.

| Method | AP$_{3D}$ IOU$\geq$ 0.7 | | | AP$_{3D}$ IOU$\geq$ 0.5 | | |
|---|---|---|---|---|---|---|
| | Easy | Mod. | Hard | Easy | Mod. | Hard |
| FCOS3D [15] | 9.55 | 5.51 | 4.78 | 34.29 | 25.78 | 23.66 |
| +Local cons. | 14.62 | 12.42 | 11.02 | 39.11 | 26.86 | 25.62 |
| +Prob. depth | 19.10 | 16.04 | 14.83 | 47.64 | 37.45 | 33.29 |
| +Depth prop. | 21.36 | 16.60 | 15.60 | 50.57 | 39.78 | 34.18 |

Table 4: Results on the KITTI test set.

| Method | AP$_{3D}$ IOU$\geq$ 0.7 | | |
|---|---|---|---|
| | Easy | Mod. | Hard |
| MonoDIS [12] | 10.37 | 7.94 | 6.40 |
| M3D-RPN [13] | 14.76 | 9.71 | 7.42 |
| MonoPair [42] | 13.04 | 9.99 | 8.65 |
| MoVi-3D [53] | 15.19 | 10.90 | 9.26 |
| RTM3D [14] | 14.41 | 10.34 | 8.77 |
| PGD (Ours) | **19.05** | **11.76** | **9.39** |

Table 5: Ablation study on nuScenes.

| Methods | mAP | mATE | mASE | mAOE | mAAE | NDS |
|---|---|---|---|---|---|---|
| FCOS3D [15] | 0.319 | 0.743 | 0.265 | 0.543 | 0.155 | 0.389 |
| +Local cons. | 0.325 | 0.721 | 0.266 | 0.546 | 0.164 | 0.393 |
| +Prob. depth | 0.339 | 0.716 | 0.265 | 0.511 | 0.163 | 0.404 |
| +Depth prop. | 0.348 | 0.701 | 0.268 | 0.452 | 0.166 | 0.415 |

Table 6: Ablation study for the depth score.

| Method | AP$_{3D}$ IOU$\geq$ 0.7 | | |
|---|---|---|---|
| | Easy | Mod. | Hard |
| Top-2 score | 20.58 | 16.30 | 14.99 |
| Norm. Entropy | 20.15 | 15.89 | 14.67 |
| 1 - Std. | 20.68 | 16.26 | 14.94 |

problem can be solved decently with enough data. At the same time, the methods using other modal data usually yield better NDS, mainly because the mAVE is smaller. The reason is that they can predict the speed of objects from continuous multi-frame point clouds or velocity measurement of Radar. In contrast, we only use the single-frame image in our experiments. So how to mine the speed information from consecutive frame images will be a direction worthy of exploring in the future. On the validation set, we compare our method with the best open-source center-based detector, CenterNet. Our method is not only much more efficient to train and inference (3 days to train the CenterNet vs. only one day to train our model with comparable performance), but also achieves better performance, especially in terms of the mAP and mAOE. On this basis, we finally achieved an improvement of about 9% on NDS. See more detailed results about depth estimation accuracy and per-class detection performance in supplementary materials.

Table 7: Ablation study for probabilistic depth.

| prop. branch | w/ direct | depth score | Easy | Mod. | Hard |
|---|---|---|---|---|---|
| ✓ | | | 14.53 | 11.93 | 10.70 |
| ✓ | ✓ | | 16.58 | 13.82 | 13.49 |
| ✓ | ✓ | ✓ | 19.10 | 16.04 | 14.83 |

Table 8: Ablation study for geometric depth.

| fusion | edge gating | cut off grad. | Easy | Mod. | Hard |
|---|---|---|---|---|---|
| ✓ | | | 18.54 | 15.44 | 13.94 |
| ✓ | ✓ | | 19.50 | 15.87 | 14.19 |
| ✓ | ✓ | ✓ | 21.36 | 16.60 | 15.60 |

## 5.4 Ablation Studies

Finally, we conduct ablation studies to validate the efficacy of our proposed key components on KITTI (Tab. 3) and nuScenes (Tab. 5). We can observe that local constraints can basically enhance the baseline, and our probabilistic and geometric depth further boost the performance significantly, especially in terms of mAP and translation error (mATE). Tab. 7 and Tab. 8 show more details of two core components for improving depth estimation with the metrics average precision under IOU$\geq$0.7. It can be seen that combining the probabilistic representation (prop. branch in the Tab. 7) with direct regression (w/ direct) and leveraging the depth score in the inference (depth score) can finally make the most of this design. For geometric depth, the basic fusion with local estimation can not bring the desirable gain. Improving the propagation graph via edge pruning and gating (edge gating) and cutting off the unexpected gradients propagation (cut off grad.) can help remove possible noises and prompt the learning more focused on the final integration, thus making the overall scheme much more effective. As for alternative implementations, we compare feasible methods of computing the depth score from the probabilistic distribution (Tab. 6). Compared to other more complicated ways, normalized entropy and standard deviation, our exploited top-2 score can achieve decent results with better efficiency. See more results about different depth division methods and detailed analyses for these two datasets from other perspectives like the Precision-Recall curve in the supplemental.

## 6 Conclusion

This paper targets the key challenge lying in monocular 3D object detection, instance depth estimation. Started from a basic adapted 3D detector, we firstly make in-depth oracle analyses. We surprisingly find that depth estimation is the dominating bottleneck for current 3D detection, especially in terms of localization. To tackle the discovered challenge, we propose a novel approach, Probabilistic and Geometric Depth (PGD), which leverages the geometric relationship in perspective to construct a graph connecting instance estimations with uncertainty and thus predicts depths more accurately. The efficacy of this solution is demonstrated on both KITTI and large-scale nuScenes datasets. In the future, we will further extend the geometric depth scheme to more general cases by relaxing the "ground" assumption via 2D height regression or ground normal estimation, and validate this pipeline on other 2D detectors. How to better leverage temporal geometry information to address the difficulty of instance depth estimation is also a promising direction worthy of further exploration.

**Acknowledgments**

This work is supported in part by Centre for Perceptual and Interactive Intelligence Limited, in part by the GRF through the Research Grants Council of Hong Kong under Grants (Nos. 14208417, 14207319 and 14203518) and ITS/431/18FX, in part by CUHK Strategic Fund and CUHK Agreement TS1712093, in part by the Shanghai Committee of Science and Technology, China (Grant No. 20DZ1100800).

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
