# OpenReview forum: "Probabilistic and Geometric Depth: Detecting Objects in Perspective"
_robot-learning.org/CoRL/2021/Conference — CoRL2021 Poster_

### Official Review · Reviewer_2epL · 2021-07-22

**Originality:** Good
**Technical Quality:** Very Good
**Clarity Of Presentation:** Excellent
**Impact:** 3

**Recommendation:**

Weak Accept: I recommend accepting the paper, but will not argue for my recommendation if the majority of other reviewers have a different opinion.

**Summary:**

This paper describes an approach to improving the performance of monocular 3D object detection systems by improving their performance in *depth estimation* specifically.  The main contributions consist of two parts:

* A quantitative study of the performance-limiting elements of a monocular 3D object detection pipeline.  The paper investigates the effect that several estimation subtasks (depth estimation, offset, size, orientation, etc.) have on overall 3D object detection performance using an *oracle analysis*: in this approach, various subsets of these *estimated* quantities are replaced with their *ground truth* values, in order to determine the sensitivity of end-to-end system performance on each of these estimation subtasks.  The results provide convincing evidence that depth estimation is by far the most influential subtask in determining overall system performance.

* Based on this insight, the paper proposes an approach for improving the performance of instance depth estimation by incorporating contextual depth cues obtained from other nearby detected objects, using a simple geometric argument based upon (i) perspective camera geometry and (ii) the assumption (reasonable in autonomous driving applications) that detected objects are supported by a common ground plane.

Both the oracle investigation and experimental evaluation of the proposed method are evaluated on two standard large-scale datasets (KITTI and nuScenes).


**Issues:**

I have no major issues (beyond the minor suggestions I mentioned in "Strengths and Weaknesses") that I would specifically like the authors to address during revision.

**Reviewer Expertise:**

Good: General knowledge of the area

**Strengths And Weaknesses:**

In my estimation the paper makes a solid contribution both scientifically and technologically.  From a scientific perspective, it sets out to investigate a fundamental and clearly practically-important question (namely: what is the “performance limiting step” in an existing monocular 3D object detector?), and devises a very simple but effective means of answering it.  [I want to specifically call out the study reported in Section 3 as particularly praiseworthy: this is one of the most well-thought-out and scientifically rigorous empirical investigations that I have seen amongst the learning-based papers that I have recently read.]  Similarly, from a technological perspective the proposed approach to improve depth estimation by leveraging the geometry of perspective projection is both elegant and (as the experiments show) practically effective.

As a result, I have only a few (relatively minor) comments / suggestions for improving the paper:

* One aspect of the proposed approach that seemed a little odd / mildly unsatisfying to me is that it appears to have at least 3 separate (i.e. independently-estimated) values relating to the estimated depth of an object instance: (i) a direct depth regression D_R, (ii) a “probabilistic depth” D_P (which appears to be modeling a belief over a set of discretized depth intervals), and (iii) a “depth confidence score” S_D (which is used as a kind of weight in the geometric fusion step).  Since the proposed method already incorporates the notion of estimating a distribution P over depth, it seems like all three of the above roles could be filled simply by P itself (i.e. without the need to introduce any additional “fudge factors”).  I would be curious to know: have the authors considered simply replacing the regression of the depth D_R with e.g. a regression of the parameters for a (continuous) *distribution* over depth? For example, a basic implementation of this might involve regressing a mean and covariance for a Gaussian distribution over depth; the mean itself could then take the place of D_R above (as a point estimate), and the covariance would presumably be used in the geometric fusion.  Potential benefits of this approach might be (i) a more streamlined / interpretable regression formulation, (ii) perhaps this framing might lead to e.g. Bayesian fusion formulas for the geometric fusion [in eqs. (7) and (8)], (iii) there would no longer be a need for discretization in the depth estimation, including the choice of binning and spacing parameters.

* As I mentioned earlier, I really liked the design of the oracle study reported in Sec. 3.  However, the figures of merit used in this study are rather complex composite *classification* scores that are designed to assess the performance of an end-to-end monocular 3D object detector, and which appear to combine (in some fashion) multiple distinct measures of performance (sometimes including arbitrary weighting parameters, thresholds, etc.).  While these are certainly informative with respect to the end-to-end downstream task (monocular 3D object detection), since ground truth is available, I would also really have liked to see a more direct assessment of the proposed approach’s performance on the depth estimation task itself.  That is: what is the direct effect of the proposed geometric fusion approach on *depth estimation*, rather than *classification score*?

* In lines 213 - 215 the paper mentions that gradients are not backpropagated through the geometric depth computation in order to “avoid influencing the learning of other components”.  However, since this computation *is* meant to be used as part of the overall pipeline, this choice seems to be at odds with the idea of joint end-to-end training.  Can the authors perhaps say a bit more about why this decision was made?

* How is lambda chosen in (2)?

* I think “D_L” in line 155 should be “D_R”


**Summary Of Recommendation:**

My recommendation is based upon the following considerations:

* The oracle study in Section 3 addresses a fundamental scientific question about the performance of existing monocular 3D object detectors (what is the performance-limiting step?) in a rigorous, thorough, and convincing manner, and arrives at a sharp and practically-useful conclusion (depth estimation).
* On the basis of this study, the paper proposes an elegant and practically-effective method for improving the performance of depth estimation by leveraging perspective geometry.

Both of these contributions are likely to be of interest to a sizable fraction of the robotics community.

---

> ### Author Response · Authors · 2021-08-18
> **Reply to Reviewer 2epL**
>
> First, thanks for the recognition of our study and methodology. Below we will respond to the raised questions and suggestions in detail.
>
> 1. The design of three separate values for depth prediction is a little odd?
>
> There are several perspectives about this question:
>
> - Clarification about the “Fusion weights $\alpha$”
>
> First, there is a minor point required to be corrected in the question. The three predictions are $D_R$, $D_P$, and the location-aware weights $\alpha$ used to integrate local estimation and geometric depth in the final step. The depth confidence scores $S_D$ can be directly derived from the depth distribution (line 153). This weight is necessary for leveraging different estimations flexibly because each estimation can be inaccurate in different contexts (please see the specific reason and analysis in the paper).
>
> - Why do we keep the $D_R$ after adopting the $D_P$?
>
> In the beginning, we just added $D_P$ instead of replacing $D_R$ for convenient comparison in the experiments. Afterward, we removed that part but observed a performance degradation, as shown in the ablation study for probabilistic depth (line 1 vs. 2 of Tab. 7). We found that although this representation itself makes the depth regression easier to learn, only combining these two methods together can achieve the best performance. It implies that these two components are complementary, and the regression branch can be responsible for regressing the residual of the probabilistic estimation (as analyzed in the supplemental material, line 172-177).
>
> - How about directly regressing a mean and covariance for a Gaussian distribution over depth?
>
> Here we discuss this potential implementation from two aspects:
>
> a. For the point estimation (each instance), regressing the mean, standard deviation, and offset for each point is a reasonable alternative (estimation = mean + std * offset). It shares a similar philosophy and is somewhat equivalent to our discretization-based method.
>
> b. For geometric fusion, estimating the covariance of any two objects from a dense prediction map is computationally expensive. In comparison, a feasible way is to integrate the depth predictions from a local patch with suitable weights, such as center-ness. Due to our center-ness being redefined with 2D Gaussian distribution, this fusion scheme can achieve a similar effect to the proposal. In our experiments, this approach does not bring a promising performance gain. We conjecture that the nearby predictions hardly contribute much to the final estimation. After all, only those pixels close to centers can extract high-quality features.
>
> In conclusion, the proposed approach is indeed a good alternative and worthy of further attempts, but it can not avoid the aforementioned necessary designs.
>
> 2. The direct assessment of the proposed method on the depth estimation?
>
> - We show the ablation in Tab. 5 in the supplemental material. Our method reduces the depth estimation absolute/relative error significantly. Besides, another important indicator is the reduction of mean Average Translation Error (mATE) in the ablation study on nuScenes (Tab. 5) of the main paper. Because the 3D-center localization heavily relies on an accurate depth (2D center localization is much easier), the reduction of mATE can strongly prove the improvement of depth estimation. We will include references for relevant content in the main paper.
>
> 3. About the reason for cutting off gradients propagated back from the geometric branch?
>
> - As the derived perspective relationship (Eqn. 6) shows, computing $d_2$ requires other estimations, such as the 2D center location, 3D size, and depth estimation. Considering so many variables/estimations are involved, it will be hard to learn them in this branch simultaneously with other branches. In addition, the ground constraint is not precisely fit in all cases. It will also bring some unnecessary noises to related learning procedures. We also empirically validate the effectiveness of this approach and show the results in the ablation study for geometric depth (line 2 and 3 of Tab. 8).
>
> 4. $\lambda$ in Eqn. (2)
>
> - $\lambda$ is basically a single data-agnostic learnable parameter. In the practical implementation, we initialize it with $10^{-5}$, and finally get a value about -1.07 (corresponding to its sigmoid response equal to 0.256) as given in the supplemental material (line 174). It means that the direct regression accounts for about 25.6\% in the preliminary point estimation for each instance.
>
> 5. $D_L$ in line 155 is indeed a typo. We will fix it with $D_R$ in the revised version.

---

### Official Review · Reviewer_Ngti · 2021-07-23

**Originality:** Good
**Technical Quality:** Very Good
**Clarity Of Presentation:** Good
**Impact:** 3

**Recommendation:**

Weak Accept: I recommend accepting the paper, but will not argue for my recommendation if the majority of other reviewers have a different opinion.

**Summary:**

The authors propose a monocular depth estimation approach which predicts probabilistic depth (via discrete buckets) for objects in a scene and then performs graph-based reasoning (along with some assumptions about ground location) to refine the single-object estimates. In this way, low uncertainty objects can improve the depth estimates for proximate higher uncertainty ones. The authors validate their approach against multiple competing methods and baselines on two autonomous driving datasets.

Update (Post Author Response):

I appreciate the authors' response to questions. Fundamentally my impression of the paper is unchanged and my initial recommendation of acceptance is unchanged.

**Issues:**

- Please elaborate on when you expect the assumptions you introduce to be violated and how much of a problem that will be.

- (Minor) Do a general editing pass for flow and clarity.

**Reviewer Expertise:**

Good: General knowledge of the area

**Strengths And Weaknesses:**

The paper proposes an interesting mechanism for injecting additional structure into the monocular depth-estimation setting. They conduct a reasonable assortment of experiments and do a nice job experimentally validating their approach.

My biggest question about this paper is how the method transfers to less structured settings. Would this work in something like a household environment where object ground locations are less consistent? Similarly, what if some objects are not on the ground (e.g. tree branches, hanging signs, etc)? The authors only report results for vehicle detection (which are very well behaved w.r.t. this manually injected inductive bias). I would have also liked to see the results of an ablation looking at only the probabilistic depth formulation (without any of the local geometric consistency or constraints).



Minor comment: Some of the prose is unnecessarily floury and the meaning can be a bit hard to parse in places. I suggest an editing pass focused on shortening sentences and simplifying language as much as possible.

**Summary Of Recommendation:**

The paper presents an interesting extension of current monocular depth estimation approaches. While some of the assumptions the authors inject seem overly suited to vehicles, the paper still provides valuable insight to others in the field and their proposed method looks promising.

I dislike the lack of granularity in some of the scores (I don't know why the two positive options for impact are <little> and <a ton>. This paper has the potential to have moderate impact if the results end up being replicated in future works by different groups. This is generally the case for most "good" papers: their impact lies somewhere in between a "3" and a "4".

---

> ### Author Response · Authors · 2021-08-18
> **Reply to Reviewer Ngti**
>
> 1. The main question is about the extension of this method to less structured scenarios.
>
> - First, we have designed some mechanisms to avoid noises from exceptional cases that violate the “ground” constraint. For example, in the edge gating scheme (Eqn. 7), we will consider the 2D distance of two objects in the image and tend to leverage the depth cues of nearby objects. For experiments, there are also unstructured samples in the nuScenes dataset, such as some cars on the side slope. The experimental results have validated that our scheme can basically handle such cases.
>
> - As we mentioned in the Conclusion Section, we will further attempt to extend the geometric relationship to more general cases by relaxing the “ground” constraint via 2D height regression or ground normal estimation in the future. Specifically, we can derive a precise (instead of an approximate) relationship by replacing the $v$ by $h^{2D}$ in Eqn. 6, i.e. $d_2=\frac{h^{2D}_1}{h^{2D}_2}d_1+\frac{f}{h^{2D}_2}(h^{3D}_1-h^{3D}_2)$. Here $h^{2D}$ refers to the 2D height between the top center and bottom center. In this way, we only need to regress an additional 2D attribute $h^{2D}$ for each instance and can avoid the “ground” constraint. Preliminary experiments have shown promising results: The extended height-based depth propagation mechanism can achieve better performance (mAP 0.318 & NDS 0.393 vs. mAP 0.305 & NDS 0.383 on nuScenes) than its counterpart (only using the single-object height-based geometry to compute depth). We will conduct experiments in more various scenarios to explore this problem further in the future.
>
> 2. Another minor question is about the improvement brought by only probabilistic representation.
>
> - We show the results in the ablation study (Tab. 3 and Tab. 5) of the main paper. It shows that this formulation not only serves as an important prerequisite for depth propagation, but also brings significant performance gains, especially for experiments on KITTI (where a more strict metric is adopted for evaluation).
>
> Finally, we will do a general editing pass for flow and clarity. Thanks for the suggestions and questions.

---

### Official Review · Reviewer_HxyZ · 2021-07-26

**Originality:** Very Good
**Technical Quality:** Excellent
**Clarity Of Presentation:** Excellent
**Impact:** 4

**Recommendation:**

Weak Accept: I recommend accepting the paper, but will not argue for my recommendation if the majority of other reviewers have a different opinion.

**Summary:**

The paper presents an extension of FCOS3D that predicts depth probabilistically and integrates it in a graph neural network to estimate relative depths across an entire scene. The resulting system achieves state of the art mono-depth-based 3D detection results on nuScenes and KITTI.

**Issues:**

See above.

**Reviewer Expertise:**

Excellent: Expert knowledge on the topic of the paper

**Strengths And Weaknesses:**

+ Easy to understand (well written)
+ simple and interesting idea with good performance
- Please add FCOS3D to T1 and T2

I like this paper. It's well written, easy to follow and has a refreshingly new take on monocular 3D detection. The paper is well executed, including good ablations and comparisons. There only minor improvements required to make this work publishable at CoRL.

Please add FCOS3D to table 1 and table 2, since PGD is already built on that baseline.

Minor:
* PointPillars [2] in T2 can work much better with the right backbone, consider citing the numbers in [50]
* "Monocular 3D detection, as an economical solution compared to conventional settings relying on binocular vision or LiDAR, has drawn increasing attention recently but still yields unsatisfactory results.": It's unclear to me why a ($50 or less if mass produced) stereo camera is more expensive than a mono-camera with a beefy GPU attached to it (LiDAR I agree is currently still more expensive)? Please comment, or rephrase.

**Summary Of Recommendation:**

I like the paper, and hope the authors incorporate the above feedback to lift the submission well above bar.

---

> ### Author Response · Authors · 2021-08-18
> **Reply to Reviewer HxyZ**
>
> Thanks for the positive feedback. Our responses to the mentioned minor questions are as below.
>
> 1. Cite higher performance of PointPillars with better backbone?
>
> - PointPillars can achieve much better performance indeed with a stronger backbone. Nevertheless, that has to sacrifice its computational efficiency significantly. We use this reported number to respect its real-time efficiency claimed in its original paper and distinguish it from other high-performance baselines (like CenterPoint). For integrity, we will consider adding its enhanced version or related remarks to our quantitative analysis.
>
> 2. Why is the monocular solution more economical than stereo?
>
> - In the stereo setting, the costs for hardware devices (two cameras) are low, but calibrating them in complex environments is inconvenient and complicated. That brings much more human resource costs compared to monocular solutions.
>
> 3. We will add FCOS3D results in Tab. 1 and 2 for a more explicit comparison.

---

### Meta-Review · Area_Chair_j3DM · 2021-08-16

**Recommendation:** Accept (Poster)
**Confidence:** 3

**Metareview:**

This paper analyzes FCOS3D by replacing upstream depth predictions with the ground-truth oracle, showing that mAP scores of 3D vehicle detections are substantially improved when depth predictions are no longer a bottleneck. This identifies depth prediction as a large bottleneck in 3D detection and its downstream semantic tasks, so the authors focus on improving depth estimation by using an uncertainty-aware refinement step of the depth.

Reviewers agree that the work is original ("Good; some advance"), has Good or Excellent clarity, and contains incremental or better contributions to the field of robotics.

I think reviewer 2epL brings up good comments & suggestions, and I'd like the authors to respond to all of them. In particular I'm also curious about why there are separate predictions D_R, D_P, S_P when they could be computed from D_P itself.

---- update 20210904
All reviewers have voted to weakly accept and the authors have addressed most reviewer comments, so I am recommending accepting this paper.

---

> ### Author Response · Authors · 2021-08-23
> **Reply to Area Chair j3DM**
>
> We thank all the reviewers and the meta reviewer for the constructive comments. We have made a minor revision for the submission according to the suggestions. The revision mainly consists of:
>
> 1. (Reviewer HxyZ) Added comments to the PointPillars cited in Table 2 and added FCOS3D (also updated PGD results (single-model) with the same training/inference setting) in Table 1 and 2 for more explicit comparisons
> 2. (Reviewer Ngti) Did overall editing for flow and clarity, especially simplified some long sentences
> 3. (Reviewer 2epL) Added a relevant reference to the results about depth accuracy in the supplemental material and fixed a minor typo of notation
> 4. Due to space limitations, we deleted the footnote about the commitment of code release. The code link will be attached in the final version.
>
> Finally, we respond to all the comments and suggestions of reviewers. Please refer to [the response to Reviewer 2epL](https://openreview.net/forum?id=bEito8UUUmf&noteId=FStf5zGqdng) for the reason of the specific design for depth prediction and a discussion about alternative methods.

---

### Decision · Program_Chairs · 2021-09-13

**Decision:**

Accept (Poster)

**Comment:**

This paper analyzes FCOS3D by replacing upstream depth predictions with the ground-truth oracle, showing that mAP scores of 3D vehicle detections are substantially improved when depth predictions are no longer a bottleneck. This identifies depth prediction as a large bottleneck in 3D detection and its downstream semantic tasks, so the authors focus on improving depth estimation by using an uncertainty-aware refinement step of the depth.

Reviewers agree that the work is original ("Good; some advance"), has Good or Excellent clarity, and contains incremental or better contributions to the field of robotics.

I think reviewer 2epL brings up good comments & suggestions, and I'd like the authors to respond to all of them. In particular I'm also curious about why there are separate predictions D_R, D_P, S_P when they could be computed from D_P itself.

---- update 20210904
All reviewers have voted to weakly accept and the authors have addressed most reviewer comments, so I am recommending accepting this paper.